# Analysis of the Choroid and Its Relationship with the Outer Retina in Patients with Diabetes Mellitus Using Binarization Techniques Based on Spectral-Domain Optical Coherence Tomography

**DOI:** 10.3390/jcm10020210

**Published:** 2021-01-08

**Authors:** Ioana Damian, Gabriela Roman, Simona Delia Nicoară

**Affiliations:** 1Department of Ophthalmology, “Iuliu Hațieganu” University of Medicine and Pharmacy, 8 V. Babes str., 400012 Cluj-Napoca, Romania; ioana.damian@umfcluj.ro; 2Medical Doctoral School 1, University of Oradea, Universitatii Str, 410087 Oradea, Romania; 3Department of Diabetes, Nutrition and Metabolic Diseases, “Iuliu Haţieganu” University of Medicine and Pharmacy, 8 V.Babes str., 400012 Cluj-Napoca, Romania; groman@umfcluj.ro; 4Diabetes Center, Emergency County Hospital Cluj, 3-5 Clinicilor Str., 400006 Cluj-Napoca, Romania; 5Clinic of Ophthalmology, Emergency County Hospital Cluj, 3-5 Clinicilor Str., 400006 Cluj-Napoca, Romania

**Keywords:** choroid parameters, diabetic retinopathy, outer retina, binarization, choroidal vascularity index, choroidal thickness, OCT

## Abstract

*(1) Background*: We aimed to reveal the relationship between the choroid and the outer retina with optical coherence tomography (OCT) in patients with diabetes mellitus (DM) with mild or no diabetic retinopathy (DR) in order to find early biomarkers for progressing retinopathy. *(2) Methods*: We performed a prospective study including 61 eyes of patients with type 1 or type 2 DM and 36 eyes of healthy controls. All subjects were imaged with Spectralis OCT. The choroid was assesseed using enhanced depth imaging OCT (EDI-OCT). Binarization of subfoveal choroidal images was done with public domain software, ImageJ (version 1.53a; National Institutes of Health, Bethesda, Maryland, USA). *(3) Results*: Luminal area, stromal area and total choroidal area were significantly decreased in diabetic patients compared to control: 0.23 ± 0.07 vs. 0.28 ± 0.08, *p* = 0.012; 0.08 ± 0.03 vs. 0.10 ± 0.04, *p* = 0.026; 0.31 ± 0.09 vs. 0.38 ± 0.11, *p* = 0.008. The thickness of retinal pigment epithelium (RPE) correlated positively with the choroidal vascularity index (CVI). The correlations between outer nuclear layer (ONL), photoreceptors (PR) and foveal choroidal thickness (FChT) were moderately negative. *(4) Conclusion*: Thicker RPE and a thinner PR layer may be assigned the role of early biomarkers signaling the conversion time to progressing retinopathy.

## 1. Introduction

The prevalence of diabetes mellitus around the world for all age groups is estimated be to 4.4% in 2030 [1]. While in 2000 there were an estimated 171 million people with diabetes mellitus (DM), it is expected that by 2030 their number will double, reaching 366 million [1]. Diabetic retinopathy (DR) is a leading contributor of all-cause blindness worldwide, with almost 35% of all patients with DM being affected by it [2]. The main retinal changes noted in patients with DM are alteration of the blood–retinal barrier with subsequent increased vascular permeability, vessel occlusion and leakage, leukostasis, mitochondrial dysfunction, neuronal swelling and apoptosis [2]. The choroid has an overwhelming importance for the outer retina, more precisely for the retinal pigment epithelium (RPE) and the photoreceptors (PRs), being responsible for their blood supply. More than that, the choroid represents the only source of metabolic exchange for the avascular fovea [2]. In addition to providing nutrients to the retina, the choroid seems to play an important role in the development of DR. Hyadat et al. reported in 1985 that in diabetic eyeschoriocapillaris, there is basement membrane thickening, capillary dropout and narrowing of the lumen and that choroidal arteries show arteriosclerotic changes [3]. These observations suggest that choroidal vasculopathy, further identified as diabetic choroidopathy (DC), might play an important role in the evolution of DR [3,4]. Fryczkowski et al. identified, by scanning electron microscopy (SEM) studies, increased tortuosity, dilation and narrowing, hypercellularity, vascular loops and microaneurysms, “drop-out” of choriocapillaris and sinus-like structure formation between choroidal lobules [5].

Indocyanine green angiography (ICG) studies in diabetic eyes reported changes like early hypofluorescent spots representing filling delays, and late hyperfluorescent spots representing intrachoroidal microvascular abnormalities [6,7]. Although it is considered the gold standard for choroidal investigation, ICG is less preferred as a monitoring tool because of its invasive nature [8]. Choroidal blood flow was found to be reduced in the foveal region in patients with DM, which was interpreted as a possible early change in DR and as a cause of tissue hypoxia and increased vascular endothelial growth factor (VEGF) levels [9,10]. There is evidence that the size and density of choroidal vessels are decreased in DM. Ferrera et al. have demonstrated a loss of intermediate and large vessels in Sattler’s and Haller’s layer in patients with DM [11]. Analysis of the choroidal vasculature anatomy showed that it is divided into small vessels that form the superficial choriocapillaris, medium-sized vessels consisting mainly of choroidal arterioles that form Sattler’s layer and deeper large vessels composed mostly of choroidal veins that form Haller’s layer [12]. It was observed that in eyes with diabetes but no DR, the choroidal vascularity index (CVI) decreases preferentially in larger choroidal veins (Haller’s layer) before medium-sized arterioles (Sattler’s layer) or smaller choriocapillaris, while the eyes with more than 5 years duration of diabetes had a lower macular CVI of Sattler’s layer when compared to the control group [12].

Optical coherence tomography (OCT) technology has changed the way the choroid is approached. Enhanced depth imaging OCT (EDI-OCT) technology producing higher resolution images with increased depth of penetration allows us to perform in vivo examinations of the cross section of the choroid. Choroidal thickness (ChT) was used for a long time as a surrogate measure for the vascularity of the choroid [13,14]. Previous studies found a thinner choroid in early DR but the thickness increased with progressive severity of DR [15,16]. The same change was noted in patients with DM without DR [17,18].

When OCT angiography was used, a decreased vessel density was found in the superficial capillary plexus (SCP), deep capillary plexus (DCP) and choriocapillaris, while the foveal avascular zone (FAZ) increased [19].

Mayumi et al. emphasized that DC means not only microangiopathy, but also diabetic neuropathy. This statement is supported by the decrease in choroidal neuronal nitric oxide synthase (nNOS) in diabetic eyes, normally found in the choroidal parasympathetic perivascular nerve fibers [20]. Lutty et al. observed that DC is an inflammatory disease in which leukocyte adhesion molecules are elevated and polymorphonuclear leukocytes are often associated with sites of vascular loss [21].

Studies regarding the ChT in different stages of DR reported confusing results like increased, decreased or not affected choroid. This variability of outcomes was attributed to a higher susceptibility of ChT to systemic or ocular factors. There was a need for another biomarker and the choroidal vascularity index (CVI) seems to have successfully replaced it. Since Branchini first described the concept of OCT in evaluating choroidal vasculature by determining the ratio of light pixels to dark pixels [14], the binarization methods, used either manually of automatically, discovered a reduction in CVI, which may indicate subclinical dysfunction within the choroid of patients with type 1 DM and type 2 DM [22,23]. Previous studies showed that photoreceptors seem to be strongly associated with DC. They are quantified as ellipsoid zone ”normalized” reflectivity, which seems to be decreased in nonproliferative diabetic retinopathy (NPDR) compared to control [24]. Moreover, choroidal damage was related to photoreceptor damage and reduced electroretinogram (ERG) [10].

In the present study, we aimed to analyze the choroidal structure in patients with DM with or without retinopathy using the OCT-based binarization method for the 1000 µm central area, knowing that the macular choroid has the maximum vascularity and the highest density of choriocapillaris [25]. We further evaluated if there was any connection between the choroidal parameters, either ChT or CVI, and the outer retinal layers, RPE, PRs or outer nuclear layer (ONL), and more specifically if a decreased or increased choroidal thickness or an altered luminal area/total choroidal area (LA/TCA) ratio quantitatively influences the outer retina at an early stage, before the patients experience any visual acuity loss. To the best of our knowledge, this approach has never been applied using the same diagnostic test.

## 2. Materials and Methods

### 2.1. Study Subjects

This prospective, single-center, comparative study adhered to the tenets of the Declaration of Helsinki. The study was approved by the Ethics Commitee belonging to Iuliu Hațieganu University of Medicine and Pharmacy (IHUMP), Cluj-Napoca, Romania, approval number 259/25.06.2018.

We included in the study 61 eyes of patients with a confirmed diagnosis of type 1 DM or type 2 DM and 36 eyes of healthy controls. All participants were recruited between July 2018 and July 2019 at the Department of Ophthalmology, Emergency County Hospital Cluj, Romania. We divided the subjects into 3 groups, group 1: healthy subjects with no DM (36 eyes), group 2: subjects with DM with no DR (NDR) (38 eyes) and group 3: subjects with DM and mild non-proliferative DR (DR) (23 eyes). Inclusion criteria for patients with DM were: (1) type 1 DM for more than 1 year or type 2 DM for more than 3 months, (2) diagnosis of no DR or mild non-proliferative diabetic retinopathy (NPDR). Exclusion criteria were: (1) DR in more advanced stages than mild NPDR, (2) diabetic macular edema (DME), (3) media opacities that could preclude appropriate OCT imaging, (4) presence of other retinal diseases such as age-related macular degeneration, vitreo-macular traction, macular hole, epiretinal membrane, retinal vein occlusion, macular scars, (5) glaucoma, optic neuropathies or neurodegenerative diseases, (6) history of ocular trauma, (7) refractive errors of more than ± 6 diopters (spherical equivalent), (8) previous ocular surgery, focal laser, panphotocoagulation laser or anti-VEGF intravitreal injection. The algorithm according to which the 3 groups were created is presented in Figure 1.

Healthy subjects were recruited from patients scheduled for routine ocular examination at the Department of Ophthalmology, Emergency County Hospital Cluj, Romania.

All subjects underwent ocular examination, including best corrected visual acuity (BCVA) evaluation with a Snellen chart, slit-lamp biomicroscopy, contact tonometry, dilated fundus examination and OCT.

We selected the eye with the best visual acuity for the analysis. If both eyes had the same BCVA, the eye with the highest OCT image quality was included in the study.

Written informed consent was obtained from the patients after explaining the required procedures and the details of the study.

ClinicalTrials.gov NCT04637217; https://www.clinicaltrials.gov/ct2/show/NCT04637217.

### 2.2. Spectral Domain-OCT (SD-OCT) Data Aquisition

All study subjects were imaged using Spectralis OCT (Heidelberg Engineering, Inc., Heidelberg, Germany).

All scans were aquired after pupil dilatation with 0.5% tropicamide and 10% phenylephrine. Images were obtained using the automated eye alignment eye tracking software (TruTrack; Heidelberg Engineering). The Spectralis ”posterior pole” scanning protocol was used: scanning area 30° × 25°, 61 horizontal raster lines centered on the fovea, to obtain volumetric retinal scans. The inbuilt Spectralis mapping software was used to perform measurements from each SD-OCT scan. Segmentation was automatically performed using Spectralis software version 6.0 to obtain the following thickness measurements: central macular thickness (CMT), outer nuclear area (ONL), retinal pigment epithelium (RPE), outer retina layers (ORLs). We extracted values only for the central 1 mm Early Treatment Diabetic Retinopathy Study (ETDRS) macular map concentric ring, defined as central thickness. The choroid was imaged using the EDI mode of SD-OCT. The macular region was scanned using a single horizontal line scan (30°) centered on the fovea, with 100 frames averaged in B-scan. Only high-quality scans, which we defined as scans with a signal strength of more than 25 db (ranging from 0 = poor to 40 = excellent) and with individual retinal layers that could be identified, were used for the analysis. All scans were performed by the same experienced operator.

The boundaries between the retinal layers are illustrated in Figure 2. We define the following parameters: central macular thickness (CMT): between internal limiting membrane (ILM) and Bruch’s membrane; outer retina: between external limiting membrane (ELM) and Bruch’s membrane, RPE layer: between the outer limit of the photoreceptor layer (PR1/2) and Bruch’s membrane; and outer nuclear layer (ONL): between the outer plexiform layer (OPL) and ELM.

In order to check out the relationship between photoreceptors and choroidal parameters, we approximated the thickness of the photoreceptor layer as follows: from the outer retina, we subtracted the RPE thickness to get the thickness of the photoreceptors’ inner and outer segments (PR 1/2); then, we added to PR 1/2 the thickness of the ONL (rod and cone cell bodies). As a result, the boundaries of the estimated photoreceptor layer are the inner limit of the RPE band and the outer limit of the OPL. We defined the inner segments of the photoreceptors (IS) as themyoid zone (MZ) and ellipsoid zone (EZ). The MZ is the hyporeflective region located between the ELM and EZ. It corresponds to the myoid portion of the inner photoreceptors’ segments. The EZ is the hyperreflective band between the MZ and outer segment (OS), previously known as the junction between the photoreceptors’ inner and outer segments; it represents the ellipsoid layer of the outer portion of the inner photoreceptors’ segments. The outer segments of the photoreceptor (OS) layer is the hyporeflective band between the EZ and the interdigitation zone (IZ). The IZ is a hyperreflective band representing the contact between the apices of the RPE cells and the outer segments of the photoreceptors; it was previously called the cone outer segment tips (COST) and rod outer segment tips (ROST).

### 2.3. Binarization of Subfoveal Choroidal EDI-OCT Images

Images were analyzed using public domain software, ImageJ (version 1.53a, https://imagej.nih.gov/ij/) [26] using the protocol as previously described by Sonoda and Agrawal with a few modifications [14,27]. First, the 1 × 1 μm image of the OCT was opened in ImageJ and the scale was set. We analyzed the non-stretched OCT scans to overcome erroneous quantification of choroidal parameters [28]. We measured a pixel length of 200 μm as given on the scale at the bottom of the OCT scan image, using the line tool. We drew a line of 1000 μm to easily delimit the area beneath the fovea. Using the ”polygon selection tool”, the total choroidal area (TCA) for the subfoveal region was selected from the outer boundary of the RPE–Bruch’s membrane layer to the choroid-sclera border, with a 500 μm width in both directions from the center of the fovea, and this represents the region of interest (ROI) that was added to the ROI manager (Figure 3B). The image was converted to an 8-bit image to allow the application of the Niblack auto local threshold tool (Figure 3C). The binarized image was re-converted to an RGB image and the luminal area (LA) was highlighted using a color threshold tool and further added to the ROI manager. In order to determine the LA within the TCA polygon, both areas from the ROI manager were selected and merged using the ”AND” operation. This third area was added to the ROI manager (Figure 3D,E). The first area from the ROI manager represents the TCA and the third one, the LA. The choroidal vascularity index (CVI), as previously defined, represents the proportion of LA to TCA. The stromal area (SA) was obtained after we substracted the LA from the TCA. The stromal area (SA), LA and TCA values were converted from μm^2^ to mm^2^.

### 2.4. Choroidal Thickness Analysis from EDI-OCT Images

The horizontal macular EDI-OCT 1x1 μm scan was uploaded in ImageJ, public domain software (version 1.53a, https://imagej.nih.gov/ij/) [26]. We first set the image scale and then we drew a horizontal 1000 μm line which served as a marker for our measurements. ChT was defined as the distance between the outer boundary of the RPE/Bruch’s membrane and the choroidal–scleral interface. Central subfoveal ChT was defined as the ChT value in the central circle of the standard ETDRS grid, which represents 1000 μm. The nasal and the temporal ChT were manually measured as 500 μm nasally and temporally to the central subfoveal point. Then we averaged the 3 values to obtain the foveal ChT (FChT), a single value that we further used in the analysis (see Figure 4).

### 2.5. Intra-rater Agreement

All the images were segmented twice, one week apart, by one grader (I.D.) to compute intra-grader reliability. The intra-rater reliability for the image binarization was measured by the absolute agreement model of the intra-class correlation coefficient (ICC). A good agreement is indicated by a value of 0.81–1.00 (see Table 1). The mean difference between the measurements was further determined by performing Bland–Altman plot analyses using MedCalc^®^ Statistical Software version 19.5.3 (MedCalc Software Ltd., Ostend, Belgium; https://www.medcalc.org; 2020) (see Figure 5). In order to increase the precision, we averaged the 2 measurements and the resulting value was further used in the study.

### 2.6. Measurement of Ocular Factors

BCVA was measured using Snellen charts and then converted to logarithm of the minimum angle of resolution (logMAR) for statistical analysis.

We used the International Clinical Disease Severity Scale DR to determine the disease severity: in eyes with ND, there were no diabetic changes in the fundus examinations, whereas eyes with DR had mild NPDR with the presence of a few microaneurysms and dot hemorrhages. All fundus examinations were performed by an ophthalmologist with experience in diabetic retinopathy.

We excluded glaucoma by assessing intraocular pressure (IOP) with slit-lamp applanation tonometry accompanied by analyzing retinal nerve fiber layer (RNFL) and ganglion cells layer (GCL) thickness on OCT scans.

### 2.7. Systemic Factors

We collected demographic data, lifestyle risk factors, medical history, medication use and glycosylated hemoglobin (HbA1C), estimated glomerular filtration rate (eGFR) and BMI values from the patients’ medical records.

### 2.8. Statistical Analysis

Statistical analysis was performed using SPSS version 20.0 (SPSS, Inc., Chicago, IL, USA). Categorical data are expressed as absolute numbers and continuous data as mean ± SD (95% confidence interval). The normality of the data distribution was confirmed via the Kolmogorov–Smirnov test. We used parametric Student’s t-tests and one-way ANOVA to compare variables with a normal distribution between groups and non-parametric Mann-Whitney tests and Kruskal–Wallis tests for variables with a non-normal distribution. A chi-square test was used to compare categorical data. Univariate and multivariate linear regression analyses were performed to assess associations between CVI (dependent variables) and ocular and systemic factors (independent variables). For multiple linear regression, factors showing significant associations in univariate analysis (*p* < 0.10) were included. Pearson correlation analyses were performed to determine relationships between ChT or CVI and ocular factors. All *p*-values were 2-sided, and *p*-values < 0.05 were considered significant.

## 3. Results

### 3.1. Demographic and Clinical Characteristics of the Study Sample

The demographic, systemic and ocular characteristics of the subjects are shown in Table 2. The mean age, gender and BMI were similar among groups. There was a significant difference regarding the duration of DM between groups: 10.31 ± 8.27 years for NDR and 14.65 ± 6.22 years for DR (*p* = 0.002). Regarding HbA1c, blood creatinine, eGFR values and treatment with insulin, no differences emerged between the two groups with DM (*p* > 0.05). BCVA was similar among all study groups (*p* > 0.05).

### 3.2. Choroidal and Retinal Parameters

When all the patients with DM were pooled as one group, the average central macular thickness (CMT) and the average foveal choroidal thickness (FChT) were similar to the control group (*p* = 0.284 and *p* = 0.180). In terms of choroidal characteristics, mean LA, SA and TCA were significantly reduced in patients with DM (*p* = 0.012, *p* = 0.026 and *p* = 0.008, respectively). The CVI was not statistically different in the DM group compared to control (*p* = 0.741) (see Table 3).

We further analyzed the patients with DM. The average CMT and mean LA, SA and TCA were decreased in each member of the DM group versus control. The same comparison between NDR patients and DR patients shows no significant difference with regard to the choroidal measurements. CMT was significantly increased in the DR group (*p* < 0.001) (see Table 4).

In Figure 6, we display mean CVI and FChT values among groups. The average CVI was increased in the NDR group and decreased in the DR group versus control. FChT was decreased in both DM groups compared to control and slightly increased in DR as compared to NDR.

In Table 5, the multiple regression model shows that increasing age is associated with a decreased CVI (*p* = 0.002). In univariate and multivariate regression, male gender was significantly associated with an increased CVI (*p* = 0.03). There were no statistically significant associations between CVI and other factors.

Univariate analysis revealed that increasing age and eGFR were significantly associated with decreased FChT and male gender. Increased LA and increased SA were significantly associated with increased FChT. However, all the associations were abolished after adjusting for potential influencing factors (such as age, gender) in the multiple regression model. LA was the only factor associated with FChT: an increased LA was significantly associated with higher FChT (*p* < 0.001) (see Table 6).

We found no significant correlation between HBP and FChT in our study groups: control group β coefficient = −0.262, *p* = 0.123, NDR group β coefficient = −0.045, *p* = 0.786, DR group β coefficient = -0.088, *p* = 0.690.

Likewise, no significant correlation was found between diabetes duration and FChT: NDR group β coefficient = 0.055, *p* = 0.744, DR group β coefficient = 0.006, *p* = 0.974.

### 3.3. Outer Retinal Layers

When we analyzed the thickness of the outer retinal layers, we found a significantly decreased ONL in NDR compared to the control group (*p* = 0.001) and in NDR compared to DR (*p* = 0.019). IS + OS was significantly decreased in NDR compared to control (*p* < 0.001) and also in DR compared to control (*p* = 0.012). PR average thickness was significantly reduced in NDR compared to control (*p* < 0.001), likewise in NDR compared to DR (*p* = 0.018).

Conversely, ONL, IS + OS and PR layers exhibited slightly increased values in the DR group compared to the NDR group (see Table 7).

In the NDR group, RPE thickness seems to be moderately positively correlated with CVI (R = 0.479 and *p* = 0.002) (see Figure 7B) and with FChT (R = 0.378 and *p* = 0.019) (see Figure 7A). In the DR group, RPE is positively strongly correlated only with CVI (R = 0.565, *p* = 0.005) (see Figure 7D). ONL thickness was moderately negatively correlated with FChT in the DR group (R = −0.447 and *p* = 0.032). The PR layer was moderately negatively correlated with FChT in the DR group (R = −0.,455 and *p* = 0.029) (see Figure 7C).

Correlations between retinal layer thickness and choroidal parameters among study groups are presented in Table 8.

The strong correlation of RPE with CVI and FChT was reconfirmed when we performed univariate linear regression (*p* = 0.002 and *p* = 0.003, respectively) (see Table 9).

## 4. Discussion

The choroid is one of the most metabolically active tissues in the human body and its contribution to the blood supply of the retina makes it extremely valuable. DC is not a new concept and the description of the choroid could be an important tool in the assessment of progressing DR. Due to its fenestrated vascular structure and to its deep location under the RPE layer, the choroid is difficult to assess, both in vitro and in vivo [14]. Previous attempts were made using fixatives, intravascular casting, flat mount preparations or scanning electron microscopy, but they were not successful in describing the proportion of vascularity in the choroid [14]. ChT was often used as a surrogate measure for the choroidal vascularity with the aim to correlate it with the stage of DR or to predict the evolution of retinopathy in patients with DM. However, the results of the previous studies are contradictory. CVI was proposed by Agrawal et al. as an indirect quantitative marker of choroidal vascularity based on EDI-OCT [8].

Since ChT was reported to decrease with age [29], we first excluded any significant difference between groups regarding the patients’ age. Increasing age was significantly associated with the reduction of CVI, in multivariate analysis, and of FChT, in univariate analysis. Gender was equally distributed in our groups and the regression analysis showed that male gender was significantly correlated with an increased CVI, as demonstrated by Tuncer et al. [30].

Duration of diabetes was 10.31 ± 8.27 years in the NDR group and 14.65 ± 6.22 years in the DR group (*p* = 0.002), but it did not correlate with FChT or CVI. This observation is in agreement with other reports, such as Torabi et al. [31], but in contradiction with Hiroaki et al. [32] or Niestrata-Ortiz et al. [33], who observed choroidal changes occurring before the development of DR and progression with increasing duration of DM despite the absence of any DR or increased retinal thickness. As Tan et al. reported, the duration and control of DM can lead to potentially confounded choroidal thickness measurements [14].

We have compared HbA1C values between groups, thinking that poor diabetic control with permanent high blood sugar levels could induce choroidal vascular damage, as previous studies reported [31,34]. We found no significant difference between NDR and DR groups concerning HbA1C values.

In order to study the association between nephropathy and choroid parameters, we analyzed eGFR values and blood creatinine which, although they were influenced by the stage of the disease (eGFR mean value was decreased and creatinine was increased in the DR group), the differences were not significant. When we further evaluated the relationship between FChT and CVI with eGFR in univariate analysis, there was a significant correlation with FChT, similarly to Kocasarac et al., which disappeared in the multivariate analysis [35].

The similar visual acuities among groups gave us the possibility to analyze the pre-clinical morphological changes of the retinal layers before any decrease in BCVA was documented.

High blood pressure had a similar distribution within our groups and it was not significantly correlated with FChT or CVI, as opposed to Ekkakwa et al. [36], who found decreased choroidal thickness that was explained by arteriolar sclerosis and vascular contraction caused by the high pressure in the choroid. However, the patients in the previous study did not have DM, which explains the difference from our findings, concidering that DM produces changes in the choroidal vessels and stroma.

We found different mean values for LA, TCA and CVI in the control group compared to published data that could be explained by the smaller measured area of 1000 µm instead of 1500 µm as in previous studies. Agrawal et al. reported mean TCA = 0.74 ± 0.21 mm^2^, mean LA = 0.49 ± 0.15 mm^2^ and mean CVI = 65.61 ± 2.33% (range, 60.07–71.27%) [37], while we found mean TCA = 0.38 ± 0.11 mm^2^, mean LA = 0.28 ± 0.08 mm^2^ and mean CVI = 74.14 ± 5.76%. Our different approach was premeditated in order to match with the retinal layer thickness values from the central ETDRS ring of 1000 µm.

When we further compared control mean values with our pooled DM group, LA, SA and TCA were significantly decreased in patients with DM, consistently with other studies [14,38,39]. Previous studies on the association between choroidal thickness and DR stage found a decreased thickness in patients with DM [12,14]. Interestingly, when we compared FChT between the NDR and DR groups, we found increased mean values in the DR group, 295.43 ± 86.83 µm vs. 269.51 ± 94.22 µm in NDR, although not statistically significant (*p* = 0.332), as found by Xu et al. and Tan et al. [14,40]. These conflicting results seem to be explained by multiple mechanisms that contribute to the choroidal thickening in different stages of DR: an initial thickening may be due to choroidal swelling secondary to DM which further induces RPE dysfunction and hyperpermeability of vessels in the choriocapillaris. A subsequent mechanism is characterized by the overexpression of cytokines activated by inflammation, oxidative stress, monocyte chemotactic protein-1, platelet-derived growth factor, VEGF, insulin-like growth factor 1, pigment epithelium-derived factor and cxc motif chemokine ligand, which are all associated with choroidal thickening. Not least, the sympathetic innervation seems to be activated in early stages of DR, which increases choroidal thickness. However, as DR progresses and hypoxia gains ground, the choroid thins, suggesting a decrease in blood flow [14,41].

Average CMT was significantly increased in DR, 290.73 ± 25.84 µm, as compared to the NDR group, 263.71 ± 22.13 µm, (*p* < 0.001), in the absence of any macular edema, either clinically or on OCT examination. This observation stands for the different stages of DR.

Mean CVI was increased in the NDR group versus control, and decreased in the DR group compared to control, but not statistically significant. Since the differences are small, they could be random, but the trend could also be real if we consider that previous studies observed CVI decreasing in parallel with DR progressing to PDR [6,37,38]. Our sample size was relatively small with regard to detecting significant differences in the initial stages of DR. The vulnerability of the submacular choroidal supply to ischemic diseases such as DR is due to numerous watershed zones of the short posterior ciliary arteries in the choroid, as demonstrated by Hayreh [25].

It is not known whether choroidal changes are the cause or the result of progressing retinopathy.

Linear regression provides us with important insights regarding FChT and CVI. LA and SA demonstrated a significant association with FChT in univariate analysis, which is preserved in multivariate analysis only for LA. This could mean that the vascular area is the predominant choroidal segment influencing the FChT, while an increase in CVI signifies either an increase in the number of blood vessels, or in the diameter of the choroidal vessels in the measured area, as Agrawal et al. explained [37].

When we compared the thickness of the outer retinal layers between groups, although there was a clear decreasing trend compared to healthy subjects, in NDR we found significant differences regarding the ONL, IS + OS and PR layer and, in DR, only for IS + OS. Further comparison between NDR and DR revealed significant differences for the ONL and PR layer. Surprisingly, mean values for ONL, IS + OS and PR layers were increased in DR versus NDR. Earlier studies found a strong connection between choriocapillaris perfusion and photoreceptor health measured by ellipsoid zone reflectivity in NPDR [24], emphasizing the dependence of the outer retinal layers on the choroid for nutrition and oxygenation [31]. The decrease in the PR layer thickness in our series, as Gella et al. also found [42], could be explained by the increased superoxide and soluble inflammatory factors produced by the PR cells themselves in response to the potentially damaging elevated glucose environment [43].

According to our results, in NDR, a moderate positive correlation was found between RPE-CVI and RPE-FChT. In DR, a strong positive correlation was identified between CVI and RPE and the correlations between ONL, PR and FChT were moderately negative. We could advance the view that increasing FChT is associated with decreased ONL and PR, which could be explained by the previously mentioned perfusion imbalance. The fact that the relationships between RPE-CVI and RPE-FChT are reconfirmed in univariate analysis entitles us to state with caution that if FChT or CVI increases, RPE layer thickness increases too. RPE cells perform phagocytosis of the shed outer segments of the photoreceptors (OS) on their apical membranes that face the photoreceptor cells. Knowing that the number of photoreceptors per RPE cell in the macula is increased and adding that phagocytosis is disturbed by hypoxia, we explain the accumulation of shed OS that is not readily engulfed. Thus, the increase in RPE thickness points to the compromised function of the outer retina [44]. We used multiple photoreceptor-related layers, such as: ONL, IS + OS and an estimated PR entire thickness to detect even smaller changes or to check if they are simultaneously affected by the choroidal damage.

Our study has some limitations. First, the sample size is relatively small. If we would have included a larger number of patients and with more severe stages of DR, the study would have been strengthened to detect differences regarding the choroid and the outer retina. Second, we did not perform a separate analysis for type 1 and type 2 DM which could have given us important insights into disease behavior. Third, we used the manual binarization method instead of a fully automated one. Fourth, we used a single foveal scan to measure FChT and to calculate the CVI. A 3D volume scan would have been more representative for macular CVI. Fifth, we did not incorporate the axial length, which might be related to choroidal thickness. These limitations will be addressed in future studies. The strengths of our study are: it is prospective; the DR group included incipient retinopathy, allowing us to search for debut changes in the choroidal parameters and in the thickness of the outer retinal layers; we adjusted for the effect of confounding factors before we confirmed for choroidal characteristics in different groups; we focused on choroidal parameters and their relationship with the thickness of the outer retinal layers. To the best of our knowledge, this approach has never been applied using the same diagnostic test.

## 5. Conclusions

Overall, LA, SA and TCA were significantly decreased in patients with DM, regardless of the presence or absence of retinopathy. Mean CVI correlated positively with RPE thickness in patients with DM. As a result of this observation, thicker RPE might signal the compromised function of the retina, being the expression of its exceeded phagocytosis capacity.

PR and ONL thickness decreases when FChT increases, in the same conditions as RPE.

These modifications were not associated with the decrease in BCVA, therefore, a thicker RPE and thinner PR layer may be assigned the role of early biomarkers signaling the conversion time to progressing retinopathy.

## Figures and Tables

**Figure 1 jcm-10-00210-f001:**
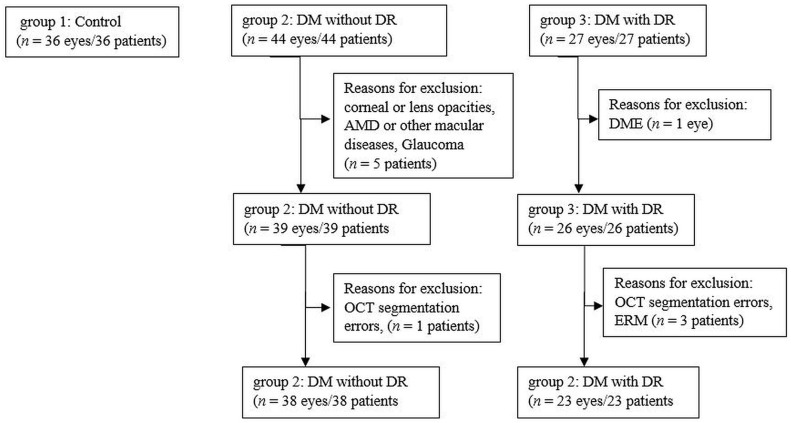
Flow diagram illustrating the study selection process. *n,* number; DM, diabetus mellitus; DR, diabetic retinopathy; AMD, age-related macular degeneration; OCT, optical coherence tomography; DME, diabetic macular edema; ERM, epiretinal membrane.

**Figure 2 jcm-10-00210-f002:**
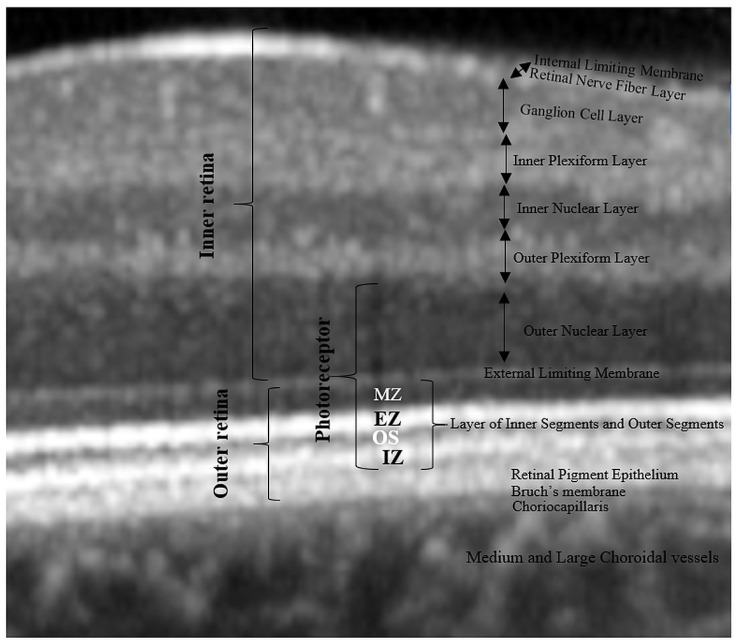
Retinal layer segmentation. MZ, myoid zone; EZ, ellipsoid zone; OS, outer segment; IZ, interdigitation zone.

**Figure 3 jcm-10-00210-f003:**
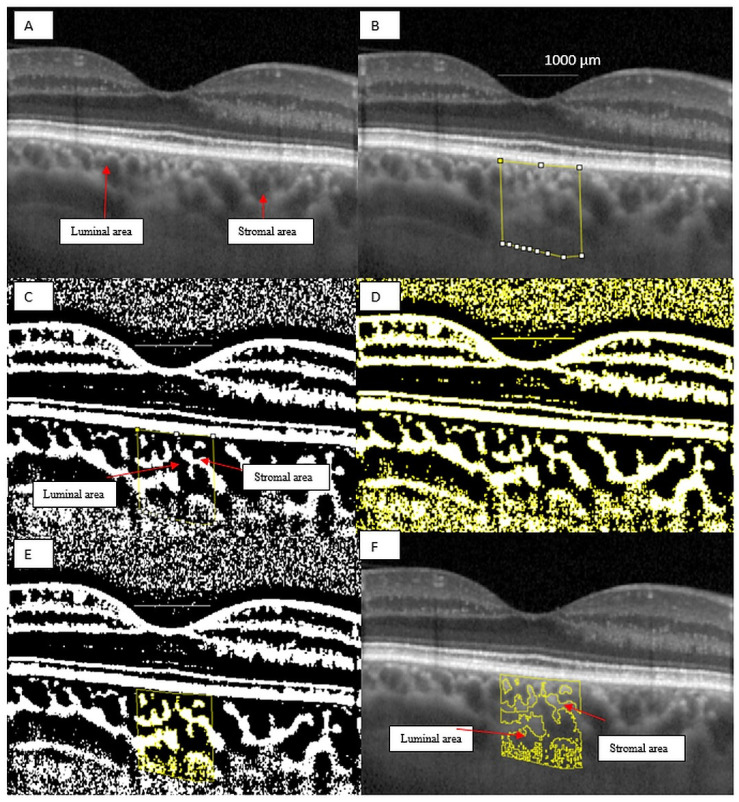
Image binarization for the choroid with normal choroidal thickness. (**A**) Original choroidal enhanced depth imaging (EDI) Spectral Domain-OCT (SD-OCT) 1 × 1 pixel * scan showing black area (luminal area, LA) and white area (stromal area, SA). (**B**) A 1 mm segment block of the subfoveal choroidal area. (**C**) The image was converted to 8-bit and Niblack auto local threshold tool was applied. (**D**) Color threshold was applied to highlight the luminal Area. (**E**) Composite image representing luminal area. (**F**) Overlay of region of interest created after image binarization was performed on the EDI-OCT image. * We used stretched 1 × 1 pixel images for illustrative purposes only, to help visualize the details of the retinal and choroidal structures.

**Figure 4 jcm-10-00210-f004:**
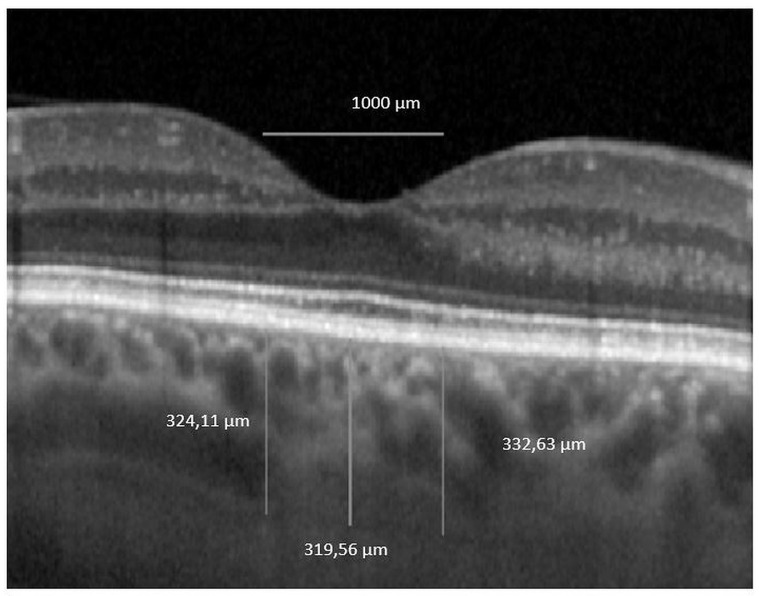
Foveal choroidal thickness. OCT 1 × 1 pixel * original image was uploaded in ImageJ software and 3 measurements were performed. * We used stretched 1 × 1 pixel images for illustrative purposes only, to help visualize the details of the retinal and choroidal structures.

**Figure 5 jcm-10-00210-f005:**
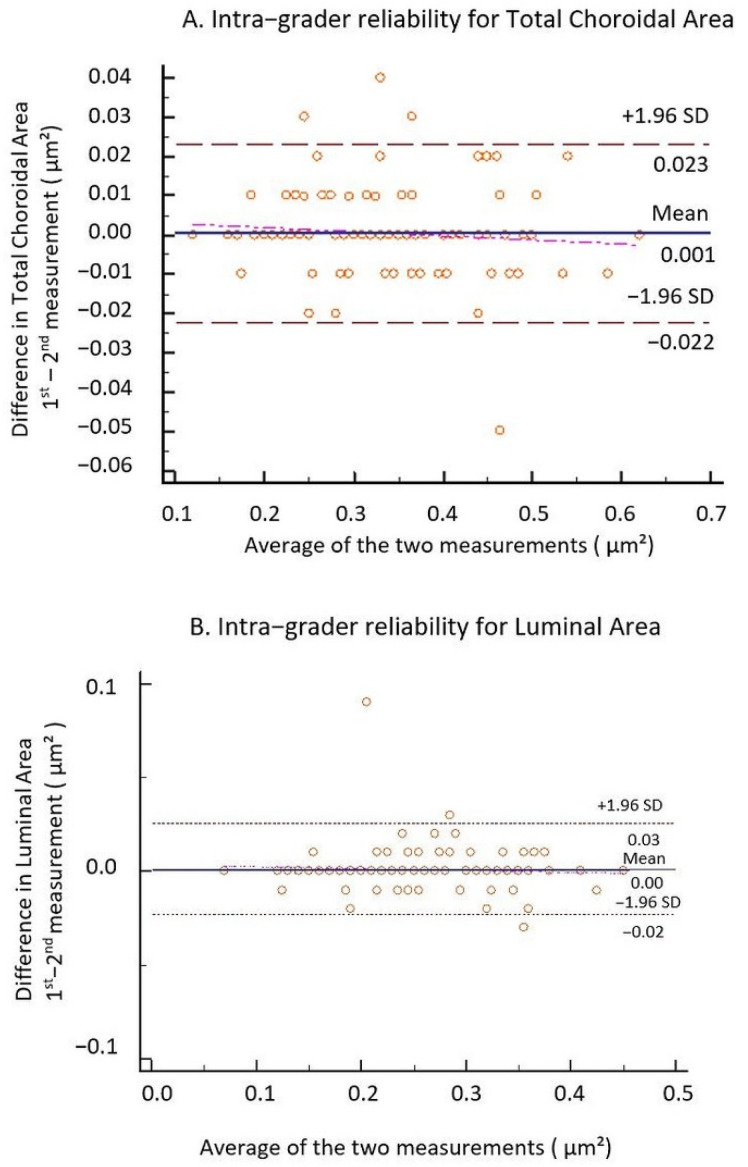
Bland–Altman plots of TCA and LA. (**A**) Intra-grader reliability for TCA and (**B**) intra-grader reliability for LA. The difference was calculated by the 1st measurement minus the 2nd measurement. Pink line represents regression line of difference between 1st and 2nd measurements.

**Figure 6 jcm-10-00210-f006:**
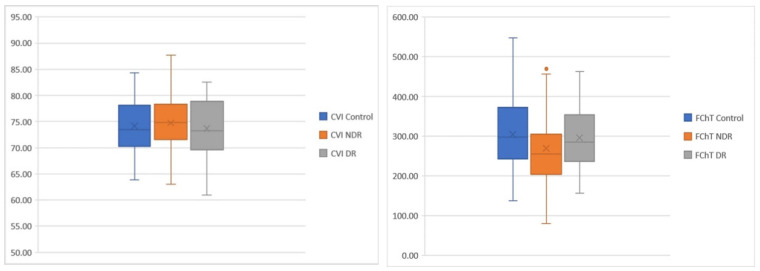
Blox plots. Differences among groups regarding CVI (**left**) and FChT **(right**).

**Figure 7 jcm-10-00210-f007:**
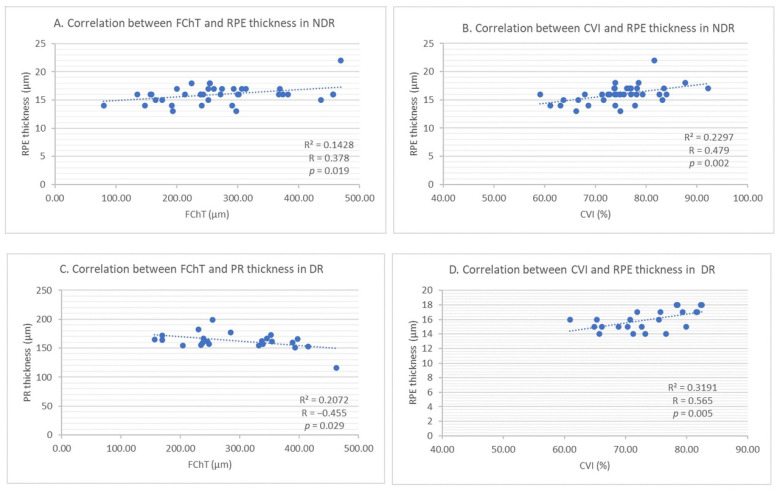
(**A**) Correlation between foveal choroidal thickness and retinal pigment epithelium thickness in NDR. (**B**) Correlation between choroidal vascularity index and retinal pigment epithelium thickness in NDR. (**C**) Correlation between foveal choroidal thickness and photoreceptors’ thickness in DR. (**D**) Correlation between choroidal vascularity index and retinal pigment epithelium thickness in DR.

**Table 1 jcm-10-00210-t001:** Intra-grader reliability assessment of choroidal parameters in 97 subjects.

ICC for TCA	ICC	95% CI	ICC for LA	ICC	95% CI
for single measures	0.9937	0.9906 to 0.9958	for single measures	0.9871	0.9807 to 0.9913
for average measures	0.9968	0.9953 to 0.9979	for average measures	0.9935	0.9903 to 0.9956

ICC, intra-class correlation coefficient; CI, confidence interval; TCA, total choroidal area; LA, luminal area.

**Table 2 jcm-10-00210-t002:** Demographic and clinical characteristics of the eyes (*n* = 97 eyes).

Characteristics	Control (*n* = 36)	NDR (*n* = 38)	DR (*n* = 23)	*p*-Value
Age, years	53.28 ± 14.19	57.08 ± 14.43	57.83 ± 11.93	0.367 ^^^
Gender: female/male, n	16/20	18/22	5/18	0.141 ^#^
BMI, kg/m^2^	26.9 ± 4.9	31.0 ± 7.3	28.5 ± 5.4	0.613 *
Diabetes duration, years	-	10.31 ± 8.27	14.65 ± 6.22	0.002 °
HbA1c, %	-	9.44 ± 2.28	9.38 ± 1.73	0.242 °
Blood creatinine, mmol/L	-	0.83 ± 0.23	1.02 ± 0.47	0.857 °
eGFR, mL/min/1.73 m^2^	-	88.73 ± 22.89	80.63 ± 28.44	0.503 ∨
Current smoking, n	8	10	9	0.046 ^#^
BCVA, logMAR	0.02 ± 0.06	0.06 ± 0.14	0.05 ± 0.08	0.575 *
Treatment with insulin, n	-	27	19	0.492 ^#^
HBP, n	18	27	12	0.141 ^#^

Data are expressed as mean ± standard deviation (95% confidence interval), unless otherwise indicated. Statistically significant *p*-values are highlighted in bold. ^^^ One-way ANOVA test, * Kruskal–Wallis test, ° Mann–Whitney test, ^#^ chi-square test, ∨ *t*-test for independent means; n, number; NDR, no diabetic retinopathy; DR, diabetic retinopathy; BMI, body mass index; HbA1c, glycated hemoglobin; eGFR, estimated glomerular filtration rate; HBP, high blood pressure; BCVA, best corrected visual acuity; LogMAR, logarithm of the minimum angle of resolution.

**Table 3 jcm-10-00210-t003:** Choroidal characteristics and central macular thickness in control group versus DM group.

Characteristics	Control (*n* = 36 eyes)	DM (*n* = 61 eyes)	*p*-Value
CMT (µm)	279 ± 20.47	273.89 ± 26.85	0.284 ^∨^
FChT (µm)	304.67 ± 88.80	279.28 ± 91.65	0.180 °
LA (mm^2^)	0.28 ± 0.08	0.23 ± 0.07	**0.012** °
SA (mm^2^)	0.10 ± 0.04	0.08 ± 0.03	**0.026** °
TCA (mm^2^)	0.38 ± 0.11	0.31 ± 0.09	**0.008** °
CVI (%)	74.14 ± 5.76	74.35 ± 6.79	0.741 °

Data are expressed as mean ± standard deviation (95% confidence interval), unless otherwise indicated. Statistically significant *p*-values are highlighted in bold; ^∨^
*t*-test for independent means, ° Mann–Whitney test; DM, diabetes mellitus; CMT, central macular thickness; FChT, foveal choroidal thickness; LA, luminal Area; SA, stromal area; TCA, total choroidal area; CVI, choroidal vascularity index.

**Table 4 jcm-10-00210-t004:** Choroidal characteristics and central macular thickness in NDR group versus DR group.

Characteristics	NDR (*n* = 38 eyes)	DR (*n* = 23 eyes)	*p*-Value
CMT (µm)	263.71 ± 22.13	290.73 ± 25.84	<**0.001** ^∨^
FChT (µm)	269.51 ± 94.22	295.43 ± 86.83	0.332 °
LA (mm^2^)	0.23 ± 0.07	0.24 ± 0.07	0.719 °
SA (mm^2^)	0.08 ± 0.03	0.09 ± 0.03	0.435 °
TCA (mm^2^)	0.31 ± 0.09	0.33 ± 0.09	0.459 °
CVI (%)	74.77 ± 7.07	73.66 ± 6.39	0.542 °

Data are expressed as mean ± standard deviation (95% confidence interval), unless otherwise indicated. Statistically significant *p* values are highlighted in bold. ^∨^
*t*-test for independent means, ° Mann–Whitney test; NDR, no diabetic retinopathy; DR, diabetic retinopathy; CMT, central macular thickness; FChT, foveal choroidal thickness; LA, luminal area; SA, stromal area; TCA, total choroidal area; CVI, choroidal vascularity index.

**Table 5 jcm-10-00210-t005:** Linear regression analyses of systemic and ocular factors associated with CVI.

	Univariate			Multivariate ^a^		
	Unstandardized β	Standardized β	*p*-Value	Unstandardized β	Standardized β	*p*-Value
*Systemic factors*						
Age, years	−0.143	−0.309	0.002	−0.132	−0.286	0.004
Gender, male	3.169	0.242	0.017	2.765	0.212	0.03
Diabetes duration, years	0.01	0.11	0.932			
BMI, kg/m^2^	−0.015	−0.14	0.889			
HBP (yes vs. no)	−0.593	−0.046	0.655			
HbA1c, %	−0.119	−0.037	0.782			
Blood creatinine, mmol/L	−3.491	−0.167	0.279			
eGFR, mL/min/1.73 m^2^	0.026	0.103	0.528			
Smoking (yes vs. no)	1.992	0.141	0.17			
*Ocular factors*						
BCVA, logMAR	−8.711	−0.14	0.17			
CMT, µm	0.019	0.074	0.471			
FChT, µm	0.002	0.027	0.791			
sFChT, µm	0.004	0.54	0.599			
TCA, mm^2^	−2.463	−0.04	0.701			

^a^ Adjusted for variables with a *p*-value < 0.10 in the univariate analysis. Statistically significant *p*-values are highlighted in bold. CVI, choroidal vascularity index; BMI, body mass index; HBP, high blood pressure; HbA1C, glycated hemoglobin; eGFR, estimated glomerular filtration rate; BCVA, best corrected visual acuity; logMAR, logarithm of the minimum angle of resolution; CMT, central macular thickness; FChT, foveal choroidal thickness; sFChT, subfoveal choroidal thickness; TCA, total choroidal area; β, regression coefficient.

**Table 6 jcm-10-00210-t006:** Linear regression analyses of systemic and ocular factors associated with FChT.

	Univariate			Multivariate ^a^		
	Unstandardized β	Standardized β	*p*-Value	Unstandardized β	Standardized β	*p*-Value
*Systemic factors*						
Age, years	−2.207	−0.335	**0.001**	0.795	0.116	0.249
Gender, male	−21.456	−0.115	0.261			
Diabetes duration, years	0.894	0.076	0.56			
BMI, kg/m^2^	−2.101	−0.145	0.156			
HBP (yes vs. no)	−30.674	−0.167	0.102			
HbA1c, %	−1.433	−0.034	0.802			
Blood creatinine mmol/L	−50.518	−0.182	0.238			
eGFR, mL/min/1.73 m^2^	1.441	0.4	**0.01**	−0.293	−0.088	0.376
Smoking (yes vs. no)	22.744	0.113	0.272			
*Ocular factors*						
BCVA, logMAR	−122.752	−0.139	0.175			
CMT, µm	−0.025	−0.007	0.947			
LA, mm^2^	1036.866	0.875	**<0.001**	865.695	0.746	**<0.001**
SA, mm^2^	1709.11	0.67	**<0.001**	469.736	0.217	0.125
CVI, mm^2^	0.77	0.054	0.599			

^a^ Adjusted for variables with a *p*-value < 0.10 in the univariate analysis. Statistically significant *p*-values are highlighted in bold. FChT, foveal choroidal thickness; BMI, body mass index; HBP, high blood pressure; HbA1C, glycated hemoglobin; eGFR, estimated glomerular filtration rate; BCVA, best corrected visual acuity; logMAR, logarithm of the minimum angle of resolution; CMT, central macular thickness; LA, luminal area; SA, stromal area; CVI, choroidal vascularity index; β, regression coefficient.

**Table 7 jcm-10-00210-t007:** Retinal layer thickness among study groups.

	Control	NDR	DR	*p*-Value	Post Hoc Analysis		
					Control vs. NDR	Control vs. DR	NDR vs. DR
RPE	16.25 ± 1.68	16 ± 1.63	15.96 ± 1.40	0.799 ^^^			
ONL	93.47 ± 11.27	84 ± 11.27	91.87 ± 14.19	**0.002** ^^^	**0.001** ^∨^	0.631 ^∨^	**0.019** ^∨^
IS + OS	72.56 ± 3.03	69.39 ± 3.61	70.39 ± 3.33	**<0.001** ^^^	**<0.001** ^∨^	**0.012** ^∨^	0.286 ^∨^
PR	166.03 ± 11.13	153.39 ± 13.12	162.26 ± 14.85	**<0.001** ^^^	**<0.001** ^∨^	0.270 ^∨^	**0.018** ^∨^

Statistically significant *p*-values are highlighted in bold. ^^^ One-way ANOVA test. ^∨^
*T*-test for independent means; NDR, no diabetic retinopathy; DR, diabetic retinopathy; RPE, retinal pigment epithelium; ONL, outer nuclear layer; IS + OS, photoreceptor inner + outer segment; PR, photoreceptor.

**Table 8 jcm-10-00210-t008:** Correlations between retinal layer thickness and choroidal parameters among study groups.

	Control				NDR				DR			
	CVI		FChT		CVI		FChT		CVI		FChT	
	R	P	R	P	R	P	R	P	R	P	R	P
RPE	−0.035	0.844	0.269	0.113	0.479	**0.002**	0.378	**0.019**	0.565	**0.005**	0.178	0.416
ONL	0.069	0.689	−0.227	0.183	0.062	0.709	−0.035	0.835	0.117	0.594	**−0.447**	**0.032**
IS + OS	−0.159	0.354	−0.037	0.830	0.308	0.060	0.170	0.306	0.312	0.146	−0.138	0.530
PR	0.026	0.878	−0.240	0.158	0.138	0.407	0.016	0.922	0.181	0.408	**−0.455**	**0.029**

Statistically significant *p*-values are highlighted in bold. NDR, no diabetic retinopathy; DR, diabetic retinopathy; CVI, choroidal vascularity index; FChT, foveal choroidal thickness; RPE, retinal pigment epithelium; ONL, outer nuclear layer; IS + OS, photoreceptor inner + outer segment; PR, photoreceptor; R, correlation coefficient; P, *p*-value.

**Table 9 jcm-10-00210-t009:** Univariate linear regression between retinal layer thickness and choroidal parameters.

	CVI			FChT		
	Standardized β	R	*p*-Value	Standardized β	R	*p*-Value
RPE	0.317	0.317	**0.002**	0.3	0.300	**0.003**
ONL	0.044	0.044	0.669	−0.133	0.133	0.193
IS + OS	0.164	0.164	0.109	0.092	0.092	0.369
PR	0.082	0.082	0.424	−0.097	0.097	0.343

Statistically significant *p*-values are highlighted in bold. CVI, choroidal vascularity index; FChT, foveal choroidal thickness; RPE, retinal pigment epithelium; ONL, outer nuclear layer; IS + OS, photoreceptor inner + outer segment; PR, photoreceptor; R, correlation coefficient.

## Data Availability

The data presented in this study are available on request from the corresponding author. The data are not publicly available due to privacy.

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
