# Peer review of "Analysis of the Choroid and Its Relationship with the Outer Retina in Patients with Diabetes Mellitus Using Binarization Techniques Based on Spectral-Domain Optical Coherence Tomography"

_jcm, 2021, doi:10.3390/jcm10020210_

Round 1
Reviewer 1 Report
Congratulation to this well written paper on choroidal parameters in patients with Diabetes mellitus. The differences are not really strong and I wonder why you think that you can see choroidal changes within the variations. Maybe you could discuss a little more in detail the two vascular parts - choriocapillaris and larger vessels. You mentioned high blood pressure in a subset - how about its influence on choroidal thickness in the different groups? How about the influence of the time diabetes is present?
Author Response
Dear Reviewer,
We are very grateful for your comments, since they helped us improve the quality of our manuscript.
We addressed all the issues that were raised and hope that our manuscript will get the acceptance for being published in this prestigious journal.
Below are the responses to the raised comments:
- “The differences are not really strong and I wonder why you think that you can see choroidal changes within the variations”
- We removed two results with no statistical significance: lines 27 – 28.
- We deleted a non-significant result regarding CVI: lines 317 – 318.
- We deleted an explanation regarding the small differences between control and DR group: line 462.
- We added a new explanation regarding choroidal changes: lines 485 – 488: “Since the differences are small, they could be random, but the trend could also be real if we consider that previous studies observed CVI decreasing in parallel with DR progression to PDR [6, 37, 38]. Our sample size was relatively small to detect significant differences in the initial stages of DR”
- “Maybe you could discuss a little more in detail the two vascular parts - choriocapillaris and larger vessels”.
We introduced an additional explanation regarding choroid anatomy: lines 80 – 87:
„Analysis of the choroidal vasculature anatomy showed that it is divided into small vessels that form the superficial choriocapillaris, medium-sized vessels consisting mainly of choroidal arterioles that form the Sattler’s layer and deeper large-sized vessels composed mostly of choroidal veins that form the Haller’s layer[12]. It was observed that in eyes with diabetes but no DR, CVI decreases preferentially in larger choroidal veins (Haller’s layer) before medium-sized arterioles (Sattler’s layer) or smaller-sized choriocapillaris, while the eyes with more than 5 years duration of diabetes had a lower macular CVI of Sattler’s layer when compared to the control group [12].”
- “You mentioned high blood pressure in a subset - how about its influence on choroidal thickness in the different groups?”
We added the correlations between high blood pressure and foveal choroidal thickness for every study group: lines 365 – 367:
“We found no significant correlation between HBP and FChT in our study groups: Control group β coefficient =-0.262, p=0.123, NDR group β coefficient= -0.045, p=0.786, DR group: β coefficient= -0.088, p=0.690”.
- “How about the influence of the time diabetes is present?”
We added the correlations between diabetes duration and foveal choroidal thickness for every study group: lines 368 – 369:
“Likewise, no significant correlation was found between diabetes duration and FChT: NDR group β coefficient=0.055, p=0.744, DR group β coefficient=0.006, p=0.974”.
We also addressed minor spelling errors throughout the manuscript.
Thanking you for the time and effort to have reviewed our work, I assure you of all my consideration.
Sincerely yours,
Prof. Dr. Simona Nicoară

Reviewer 2 Report
This is a valuable study assessin parameters of vessels of the retina and choroid in diabetic patients with and without diabetic retinopathy. Image binarization of the choroid was done. There are some limitations of the study, but they have been presented in the discussion chapter.
However, the conclusion in the abstract is not clear: “Thicker RPE and FChT point to the conversion time to progressing retinopathy and may be assigned the role of early biomarkers for assessing diabetic retinopathy.” I think some words are missing, the conclusion should be modified according to the obtained results.
Minor remark is that the description of statistical analysis should not be included in the abstract.
Moreover, the manuscript should be corrected by professional translator or native speaker.
Author Response
Dear Reviewer,
We are very grateful for your comments, since they helped us improve the quality of our manuscript.
We addressed all the issues that were raised and hope that our manuscript will get the acceptance for being published in this prestigious journal.
Below are the responses to the raised comments:
- However, the conclusion in the abstract is not clear: “Thicker RPE and FChT point to the conversion time to progressing retinopathy and may be assigned the role of early biomarkers for assessing diabetic retinopathy.” I think some words are missing, the conclusion should be modified according to the obtained results.
We removed the conclusion and replace it with : „Thicker RPE and thinner PR layer may be assigned the role of early biomarkers signaling the conversion time to progressing retinopathy” (lines 29 – 30).
We adjusted the conclusion section: lines 544 – 551.
- Minor remark is that the description of statistical analysis should not be included in the abstract.
We removed the following sentence from the abstract:,, Statistical analysis was performed using SPSS version 20.0.” (line 23)
We also addressed minor spelling errors throughout the manuscript.
Thanking you for the time and effort to have reviewed our work, I assure you of all my consideration.
Sincerely yours,
Prof. Dr. Simona Nicoară
